# The Link between ADHD Symptoms and Antisocial Behavior: The Moderating Role of the Protective Factor Sense of Coherence

**DOI:** 10.3390/brainsci12101336

**Published:** 2022-10-03

**Authors:** Haym Dayan, Mona Khoury-Kassabri, Yehuda Pollak

**Affiliations:** 1Paul Baerwald School of Social Work and Social Welfare, The Hebrew University of Jerusalem, Jerusalem 9190501, Israel; 2Seymour Fox School of Education, The Hebrew University of Jerusalem, Jerusalem 9190501, Israel

**Keywords:** ADHD, antisocial behavior, sense of coherence, protective factors

## Abstract

Numerous studies have established the link between ADHD and antisocial behavior, one of the most serious functional impairments caused by the disorder. However, research on protective factors that mitigate this link is still lacking. The Salutogenic Model of Health offers the “Sense of Coherence” (SOC), establishing that individuals who see their lives as logical, meaningful, and manageable are more resistant to various risk factors and diseases. The present study examines for the first time whether SOC is also a protective factor against different ADHD-related types of antisocial behaviors (severe/mild violent behavior, verbal violence, property crimes, public disorder, and drug abuse). A total of 3180 participants aged 15–50 completed online questionnaires assessing the level of ADHD symptoms, antisocial behaviors, and SOC. Structural equation modeling was applied to examine the research hypothesis. An interaction between ADHD symptoms and SOC was found in predicting each type of antisocial behavior (beta = −0.06–−0.17, *p* < 0.01). The link between ADHD symptoms and antisocial behavior was significantly weaker for high than low SOC participants, regardless of age group. The current study found that people with high SOC are protected against the effect of ADHD symptoms on one of the most serious functional impairments, antisocial behavior. These findings suggest that SOC is a protective factor from the adverse effects of ADHD symptoms, justifying further prospective and intervention studies.

## 1. Introduction

### 1.1. The Link between ADHD and Antisocial Behavior: The Moderating Role of the Protective Factor Sense of Coherence

ADHD is a common neurodevelopmental condition defined by symptoms of inattention, hyperactivity, and impulsivity, accompanied by academic, occupational, or social functional impairment [1]. The functional impairment that this study focuses on is antisocial behavior. This term refers to a range of antisocial behaviors, both criminal and noncriminal (such as verbal aggression) [2].

### 1.2. ADHD and Antisocial Behavior

Numerous studies have indicated that ADHD is associated with antisocial behavior and delinquent acts [3]. The results of these studies demonstrated that, in comparison to individuals without ADHD, adolescents, young adults, and middle-aged adults with ADHD tend to be more involved in the criminal justice system, are arrested and convicted at an earlier age, and have an increased risk of conviction, incarceration, and criminal recidivism [4].

The prevalence of ADHD among incarcerated people is high and estimated at around 25–45%, varying depending on the measurement method [5]. Longitudinal studies that examined childhood ADHD found that it predicts a higher prevalence of antisocial behavior in adulthood [6]. In the general population, levels of ADHD symptoms predict antisocial behavior even after controlling for additional risk factors [7]. Similarly, the association between ADHD and antisocial behavior remained significant after controlling for a wide range of psychosocial variables [4,5]. It seems, therefore, that there is a clear correlation between ADHD and antisocial behavior. Still, most people with ADHD do not engage in antisocial behavior, suggesting that other factors influence the relationship. Therefore, we suggest looking for a factor that mitigates the link between ADHD symptoms and antisocial behavior.

### 1.3. Interaction in the Prediction of Antisocial Behavior

Various variables that interact with ADHD symptoms in the prediction of antisocial behavior have been examined in the research literature. For example, it was found that targeted drug therapy for ADHD decreases antisocial behavior by 30–40% [3,4]. Other studies reported an interaction between ADHD and variables such as substance use and past arrest in predicting delinquent behavior [8,9,10]. Notably, these moderating variables are in themselves on the spectrum of antisocial behavior. Therefore, to understand how ADHD leads to antisocial behavior, it is crucial to find other factors that do not constitute a type of antisocial behavior themselves.

Several studies focused on religiosity as a potential moderator, yielding inconsistent results. In one study, religiosity increased the risk of antisocial behavior such that, for religious people, ADHD was a weaker predictor of antisocial behavior. In another study, high ADHD symptoms combined with high religious participation predicted increased levels of antisocial behavior [11]. A prospective study, which examined the effect of childhood ADHD on adolescent antisocial behavior, found that a good parent–child relationship moderated and decreased the risk of antisocial behavior. Thus, among 15-year-old adolescents whose parents showed a more considerable decrease in parental involvement, increases in hyperactivity/impulsivity predicted higher levels of antisocial behaviors.

In contrast, among adolescents whose parents showed a more considerable or no decrease in parental involvement, increases in hyperactivity/impulsivity symptoms did not predict antisocial behaviors [12]. Another study found positive parenting to moderate the relation between ADHD symptoms and positive alcohol expectancies among children in grades 4–12 [13]. ADHD symptoms predicted higher positive alcohol expectancies at high levels of positive parenting, whereas, at low levels of positive parenting, ADHD symptoms predicted lower positive alcohol expectancies. As alcohol expectancies are highly related to alcohol use, this moderation effect was not in the expected direction, yet it shows that contextual characteristics may affect the relation between ADHD symptoms and antisocial behaviors [14]. Furthermore, it was found that an agreeable personality moderated the relation between ADHD symptoms and antisocial behavior. Specifically, among adults with a childhood history of ADHD symptoms and conduct problems, higher levels of agreeableness weakened the association between inattention and level of criminal involvement [15].

To summarize, a literature review demonstrates the paucity of research on psychosocial factors that do not constitute antisocial behavior by themselves but moderate the relation between ADHD symptoms and antisocial behavior. The present study aims to fill in this gap.

### 1.4. Sense of Coherence

The Salutogenic Model of Health focuses on health promotion, setting out against what is termed the “pathogenic approach”: the assumption that the basic human condition is health, hence focusing on risk factors involved in disease generation [16,17,18]. On the other hand, the Salutogenic Model assumes that health is under constant threat by various risk factors, thus focusing on factors contributing to defense mechanisms, which help to generate health [18]. Analogously, other researchers claimed that ADHD symptoms exist on a spectrum and do not necessarily entail functional impairment [19,20] and that the severity of both symptoms and functional impairment can be recompensated by various protective factors [21,22]. Therefore, it is vital to study the protective factors that decrease various risks associated with ADHD symptoms.

The Salutogenic Model of Health offers the concept of the “Sense of Coherence” (SOC) as a protective factor. SOC consists of three components (usually treated as a unified concept): *comprehensibility* (the cognitive component)—the degree to which one perceives the world as logical, consistent, and predictable; *meaningfulness* (the emotional component)—the degree to which one sees her or his life as meaningful and worthy of effort; and *manageability* (the behavioral component)—the degree to which one views oneself as competent and capable of influencing reality [17,23]. The theory contends that individuals with high SOC, who see their lives as logical, meaningful, and manageable, are more resistant to various risk factors.

Numerous studies highlighted the relation between SOC and normative behavior [24]. For example, it was found that poor SOC was associated with a higher level of criminal offenses in young males and with recidivism [25,26]. High SOC was associated with decreased antisocial behavior, such as smoking, alcohol consumption, and violent behavior [27,28]. Moreover, a study that was conducted on 5000 participants showed that patterns of substance use in the peer group predicted substance use of adolescents better in subjects with low SOC than in subjects with high SOC [10], suggesting that coherence is a moderator of the link between risk factors and antisocial behavior.

Only a few studies examined SOC in the context of ADHD. A longitudinal study demonstrated both an association and an interaction between SOC and ADHD. Thus, 16-year-old adolescents with ADHD reported lower SOC. Moreover, ADHD scores at age 16 predicted ADHD scores at age 21 more strongly for people with lower SOC [29]. It was also found that the high SOC was positively related to life satisfaction in ADHD students [30].

To summarize, the literature suggests that high SOC constitutes a protective factor against antisocial behavior. Furthermore, it may moderate various risk factors, including ADHD symptoms, and adverse outcomes, including antisocial behavior. Therefore, the research hypothesis is that the association between ADHD symptoms and antisocial behavior will be moderated by the level of coherence and reduced among subjects with high SOC.

## 2. Materials and Methods

### 2.1. Participants and Procedure

The study was authorized by the Hebrew University School of Social Work and Social Welfare’s ethics committee. The participants were recruited by convenience sampling, using an online questionnaire distributed on social networks (Whatsapp, Instagram, Facebook, and emails) in Israel in October–November 2020. We explained that our research goal was to understand individuals’ thoughts, feelings, behaviors, and coping strategies in various life situations. For those interested in participating, we offered a lottery of purchase tickets in a separate link (the winning ratio was 1:50 and the ticket value was about USD 12). We also asked the participants to answer the questionnaires and further distribute it. A total of 3415 participants between the ages of 15 and 50 responded to the questionnaire. A total of 235 participants were excluded since they did not answer any questionnaires or answered the questionnaires randomly, resulting in 3180 that were included in the statistical analyses. Participants were asked to complete scales that measure behavior, emotions, and perceptions in different life domains. Participants were provided information regarding the study procedures and how their privacy and confidentiality will be secured.

### 2.2. Measures

Antisocial behavior was measured using a scale based on the Self-Report Antisocial behavior (SRD) [31]. In its full version, the scale includes 47 items, which describe illegal or non-normative actions, and the respondent is requested to note the number of times the actions were performed in the last year. Internal consistency of the original scale was high (alpha = 0.91) [32]. The scale was translated to Hebrew using the back-translation method; the internal consistency of the Hebrew version was high (Cronbach’s α = 0.908) [33]. The scale was shortened and adapted so it included a wide range of ages. The adapted version of the scale consisted of 23 items, which referred to the following measures: four types of violent crimes: serious physical assault (4 items, *ω* = 0.729), mild physical assault (4 items, *ω* = 0.769), verbal assault (4 items, *ω* = 0.790), and indirect violence (3 items, *ω* = 0.559)); property crimes (5 items, *ω* = 0.784); and crimes against the public order (3 items, *ω* = 0.738). As the internal consistency of the indirect violence scale was poor, this scale was not further analyzed.

Drug use was measured using a scale developed by Johnston, O’Malley, and Bachman (1995), widely used in Hebrew translation in Israel [34]. Participants were asked to indicate their substance use in the last year. Specifically, they rated their use of cannabis (2 items), alcohol (2 items), and cigarette smoking (1 item) on a 7-point Likert scale (0 = never to 5 = 30 times or more). The internal consistency of the scale was acceptable (*ω* = 0.756).

The Hebrew version of the Adult ADHD Self-Report Scale (ASRS-v1.1) was used to measure the level of ADHD symptoms [35]. A total ASRS score was given by averaging all 18 items. In order to measure positive attention and impulse regulation behaviors in the general population, the wording of the 18 ADHD symptoms was phrased positively, in a similar manner to the SWAN questionnaire [36]. The internal consistency of the scale was good (*ω* = 0.887). The convergent validity of the scale was confirmed by a strong correlation (r = 0.655) with the Hyperactivity Problems subscale of the Strengths and Difficulties Questionnaire (see below).

A 13-item version of the SOC was used. The original scale has a high consistency and validity, and the internal consistency in the current study was good as well (*ω* = 0.888) [16,17]. The different components of the scale were analyzed as one variable [23].

Emotional problems were measured by the Strengths and Difficulties Questionnaire, which measures emotional and social adaptation (SDQ) [37]. The questionnaire consists of 25 items yielding five subscales. We used the emotional problems subscale as a covariate in the present study. We also used the Hyperactivity Problems scales for testing the convergent validity of the adapted ADHD symptoms scale. The reliability and validity of the scale have been established both for children and adults [38,39].

The participants filled out a socio-demographic questionnaire that included variables related to antisocial behavior and that were comparable between adolescents and adults: age, gender, and religiosity. Religiosity was measured by the question, “what is the level of your religiosity from 0 to 6? (0 = not religious, 6 = very religious)”. Age was categorized as late adolescence (15–18 years), young adulthood (19–30 years), and mid-adulthood (31–50 years).

### 2.3. Analytic Approach

The overall score of each scale was obtained by averaging the items of each respondent. We computed descriptive statistics of the dependent, independent, and covariates and the first-order correlations among the variables. We applied a structural equation modeling (SEM) using AMOS 26 [40] with a maximum-likelihood estimation method to examine the research hypotheses. SEM has several advantages over a series of regressions (e.g., PROCESS) as it can simultaneously test multiple hypotheses regarding multiple dependent variables in a single model and, thus, avoid alpha inflation. Another advantage of SEM is its ability to treat missing data using the full information maximum likelihood. We, therefore, modeled ADHD symptoms as a predictor of the six facets of antisocial behavior, with coherence as a moderator. Age, gender, religiousness, and emotional symptoms were modeled as covariates. Prior to analysis, ADHD symptoms and coherence were centered to their means to ensure a nonbiased interpretation of the results [41]. The six dependent variables were allowed to covariate in order to account for their potential associations. We tested the significance of the model’s effects using a p-value approach accompanied by a 95% CI based on 20,000 bias-corrected and accelerated bootstrap samples, which are more robust to violation of distributional assumptions. Significant ADHD symptoms by coherence interactions were probed into the conditional effects (i.e., simple slopes) of ADHD symptoms on antisocial behavior at high (mean + 1SD) and low (mean—1SD) levels of coherence. To further explore whether the moderation effect of coherence is stable across age groups, we used a multi-group analysis in SEM. Specifically, we contrasted two models, one in which the moderation effect was freely estimated in the two age groups and a second model in which the moderation effects in the age groups were constrained to equality. A significant chi-square difference between the two models (free vs. constrained) indicates that age further modifies the moderation effect of coherence. In contrast, a nonsignificant chi-square difference suggests that the moderation effect of coherence is equivalent across age groups.

## 3. Results

### 3.1. Descriptive Statistics

The means, standard deviations, medians, and minimum–maximum values of the demographic characteristics, ADHD symptoms, coherence, and antisocial behavior types are presented in Table 1. In total, 54.5% were women (N = 1614), 89.7% identified as Jewish, and 87.3% were born in Israel. Age groups consisted of 1125 late adolescents (38%), 925 young adults (31.3%), and 910 mid-adults (30.7%). The average religiosity level (on a 0 to 6 scale, see below) was 2.77 (SD = 1.44), and the average emotional problem score (on a 1 to 3 scale, see below) was 1.56 (SD = 0.48). Notably, 10.2% of the participants reported a diagnosis of ADHD.

### 3.2. Correlations

Spearman’s rho correlation coefficients were computed between demographic variables and ADHD symptoms, antisocial behavior types, and coherence (see Appendix A). ADHD symptoms were negatively correlated with age and religiosity and positively correlated with emotional symptoms. All antisocial behavior types were negatively correlated with age, gender, and religiosity (except for mild physical antisocial behavior, which was not significantly correlated with religiosity). Coherence was positively correlated with age and religiosity and negatively correlated with emotional symptoms and was higher in men than in women.

Spearman’s rho correlation coefficients were also computed between ADHD, antisocial behavior types, and coherence (Table 2). ADHD symptoms were positively correlated with all types of antisocial behavior (severe physical, mild physical, verbal, property, public order, and drugs). The Bonferroni method was used to correct for multiple analyses. Coherence was negatively correlated with ADHD symptoms and all types of antisocial behavior. Pearson correlations were also computed, resulting in a similar pattern (see Appendix A).

### 3.3. Moderation Analysis

We used SEM to conduct moderation analyses on the six facets of antisocial behavior. To allow an easier-to-follow presentation of the complex tested model (i.e., 36 main effects, six interactions, and six dependent variables), we herein present the results per each of the six dependent variables.

The model accounted for 15% of the severe physical antisocial behavior variance. As shown in Table 3, young age, male gender, lower emotional symptoms, higher ADHD symptoms, and lower coherence were significant predictors of severe physical antisocial behavior. In addition, ADHD symptoms by coherence interaction was found to be significant. Probing the interaction into conditional effects, as presented at the bottom of Table 3 and Figure 1, indicates that ADHD symptoms predicted severe physical antisocial behavior when coherence was low but not when coherence was high.

With regard to mild physical antisocial behavior, the model accounted for 20% of the variance. As shown in Table 4, young age, male gender, religiosity, lower emotional symptoms, higher ADHD symptoms, and lower coherence were significant predictors of mild physical antisocial behavior. As expected, an ADHD symptoms by coherence interaction was found to be significant. As shown at the bottom of Table 4 and Figure 2, ADHD symptoms predicted antisocial property behavior when coherence was low but not when coherence was high.

The SEM model accounted for 22% of the variance in verbal antisocial behavior. As shown in Table 5, young age, male gender, lower religiosity levels, lower emotional symptoms, higher ADHD symptoms, and lower coherence were significant predictors of verbal antisocial behavior. In addition, ADHD symptoms by coherence interaction was found to be significant. Probing the interaction into the conditional effects, as presented at the bottom of Table 5 and Figure 3, ADHD symptoms predicted verbal antisocial behavior more strongly when coherence was low than when coherence was high.

The tested model accounted for 12% of the variance in property crimes. As shown in Table 6, young age, male gender, lower religiosity levels, lower emotional symptoms, higher ADHD symptoms, and lower coherence were significant predictors of property crimes. As expected, a significant ADHD symptoms by coherence interaction was found. As shown at the bottom of Table 6 and Figure 4, ADHD symptoms predicted property crimes when coherence was low but not when high.

With regard to public order antisocial behavior, the model accounted for 20% of the variance. As shown in Table 7, young age, male gender, lower religiosity levels, lower emotional symptoms, higher ADHD symptoms, and lower coherence were significant predictors of public order antisocial behavior. In addition, ADHD symptoms by coherence interaction was found to be significant. Probing the interaction into conditional effects, as presented at the bottom of Table 7 and Figure 5, indicates that ADHD symptoms predicted public order antisocial behavior more strongly when coherence was low than high.

The SEM model accounted for 18% of the drug’s antisocial behavior variance. As shown in Table 8, older age, male gender, lower religiosity levels, lower emotional symptoms, higher ADHD symptoms, and lower coherence were significant predictors of drug use. In addition, ADHD symptoms by coherence interaction was found to be significant. As shown at the bottom of Table 8 and Figure 6, ADHD symptoms predicted drug use when coherence was low but not when high.

### 3.4. Age as a Moderator

We tested an additional model in which we added the age group as a moderating variable to examine whether the ADHD symptoms by coherence interaction is stable across age groups. Using a multigroup approach in SEM, we contrasted the fit indices of a free model versus a constrained model in which the ADHD symptoms by coherence interaction was forced to equality across age groups. Results indicated that the difference between the model was not significant (ΔChi-square = 9.13, df = 6, *p* = 0.168). This suggests that the age group does not significantly modify the moderation effect of coherence and that the reported conditional effects of ADHD symptoms on the six facets of antisocial behavior are similar across age groups.

## 4. Discussion

The present study examined the interaction between ADHD symptoms and the sense of coherence (SOC) in predicting various types of antisocial behavior. The research hypothesis was that SOC, which constitutes a protective factor in coping with diseases and neutralizing other risk factors, will also be a protective factor moderating the antisocial behavior risk that stems from ADHD symptoms [42].

This study examined several types of antisocial behavior and found that SOC moderated the relation between ADHD symptoms and antisocial behavior; for people with a strong SOC, the relation between ADHD symptoms and antisocial behavior decreased; in certain types of antisocial behavior, it was neutralized; and, in some cases, a negative association was found between ADHD symptoms and antisocial behavior. The research findings suggest that SOC constitutes a protective factor for the antisocial behavior risk of ADHD symptoms, similar to its ability to moderate another kind of risk for antisocial behavior, namely, peer influence. Specifically, the peer group’s patterns of substance use predicted adolescents’ substance use better in subjects with low SOC than in subjects with high SOC [10].

Assuming that SOC indeed decreases the levels of antisocial behavior, the generalizability question arises, i.e., whether SOC specifically protects against antisocial behavior or constitutes a general protective factor that decreases all functional impairments related to ADHD symptoms, including the impairment in well-being. The moderating role of coherence in protecting against other risks to well-being has been examined in many studies [42]. For example, a study investigating the relationship between work characteristics and well-being found that SOC has a moderating role in the relationships between specific risk factors and well-being. In another study, the association between social relations at work and well-being was stronger among subjects with a very low SOC, whereas these associations were less critical in determining well-being in subjects with a stronger SOC [43]. This question of moderating the risk of ADHD symptoms to other functional impairments is especially pertinent to the present study’s findings, since, in this study, ADHD symptoms were measured by self-report without an ADHD diagnosis. As ADHD diagnosis relies on the combination of ADHD symptoms and self-report of functional impairment, it is possible that participants who reported high SOC are those whose symptoms did not cause functional impairment and would not meet the criteria of ADHD diagnosis.

The main contribution of this study is showing that ADHD symptoms constitute a risk factor for antisocial behavior only when combined with further factors. This approach corresponds to the WHO’s International Classification of Functioning, Disability and Health (ICF) manual [44]. The ICF distinguishes between several concepts: *impairments*—problems in body function and structure, such as significant deviation or loss; *activity limitation*—difficulties an individual may have in executing activities; and *participation restriction*—problems an individual may experience in involvement in life situations. The individual’s functioning and health consist of the interactions between these three components with personal and developmental factors [44]. The ICF model was implemented in previous studies of ADHD [22,45,46]. The present study adds to the existing literature by implementing the ICF model in the context of ADHD symptoms and antisocial behavior.

Antisocial behavior can be defined as one of the central functional impairments associated with ADHD, which creates a participation restriction in normative society. The findings call for the application of the ICF model in the domain of ADHD and antisocial behavior, suggesting that the antisocial behavior stemming from ADHD is a product of the interaction between ADHD symptoms and a personal factor (SOC). This finding is consistent with the growing trend in the research literature calling for examining resilience factors for the risks for functional impairments associated with ADHD [12,21,22,45,46,47,48,49,50,51].

## 5. Limitations

Notably, since this research is cross-sectional, one cannot infer causality, and this issue needs to be examined using longitudinal and interventional designs. Findings of previous longitudinal studies indicate that SOC constitutes a protective factor for other outcome variables. For example, it was found that a strong SOC moderates the risk for long-term persistence of ADHD [29]. In addition, the phrasing of the items in the present study may also suggest that higher levels of SOC preceded the decrease in antisocial behavior. While participants were requested to refer to a general SOC in their life, in the antisocial behavior scale (SRD), they were requested to refer only to the last year.

The measuring of antisocial behavior relied on participants’ self-report without collateral support from other sources, such as partners’ reports or criminal records. However, it allows referring to antisocial behaviors that are not documented in criminal records. Notably, criminal records might contain a certain rate of false convictions due to false confessions, which are more prevalent among individuals with ADHD [52].

Another limitation is that the study examined ADHD symptoms in the past six months. Referring only to adulthood ADHD symptoms might supply a partial picture, since ADHD symptoms tend to be less conspicuous in adults [6]. Additionally, measuring ADHD symptoms using only self-report tends to lessen the actual level of ADHD symptoms [53]. Furthermore, in some studies, Adult ADHD Self-Report Scale had low specificity [54,55]. It is important to note that the research focuses on ADHD symptoms in the general population and not on diagnosed ADHD. Further research is needed to determine whether these correlations exist also in the diagnosed population. Finally, the study was conducted during a global pandemic, which might have influenced the level and severity of antisocial behavior [56].

## 6. Conclusions

Our study highlights the SOC as a protective factor that can be influenced by therapeutic intervention, similarly to what has been performed in the context of other disorders [57,58,59,60,61]. In light of the research findings, there is a need to explore the ICF model in the context of ADHD, using longitudinal studies, based on the understanding that functional impairments, such as antisocial behavior, are possible products of a relation between ADHD symptoms and other protective factors. Moreover, an interventional study is also needed to explore new methods of strengthening SOC among individuals with ADHD.

## Figures and Tables

**Figure 1 brainsci-12-01336-f001:**
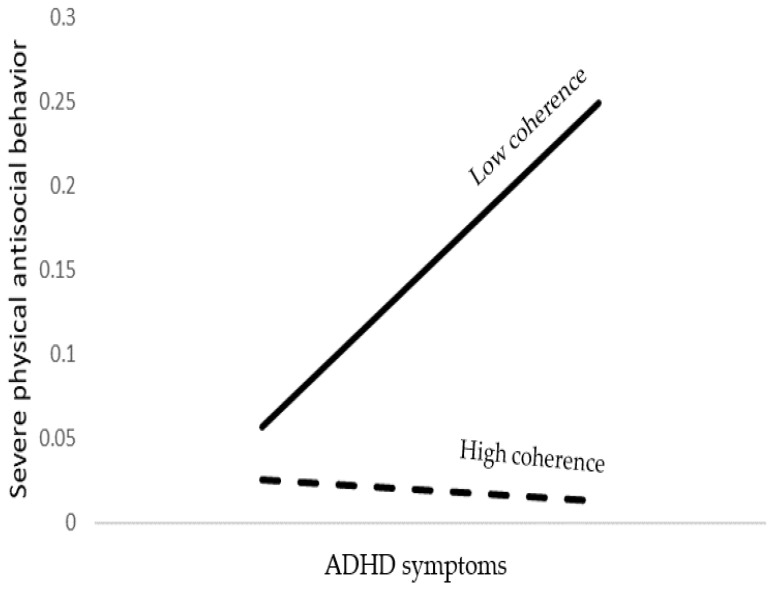
The association between ADHD symptoms and severe physical antisocial behavior as moderated by coherence.

**Figure 2 brainsci-12-01336-f002:**
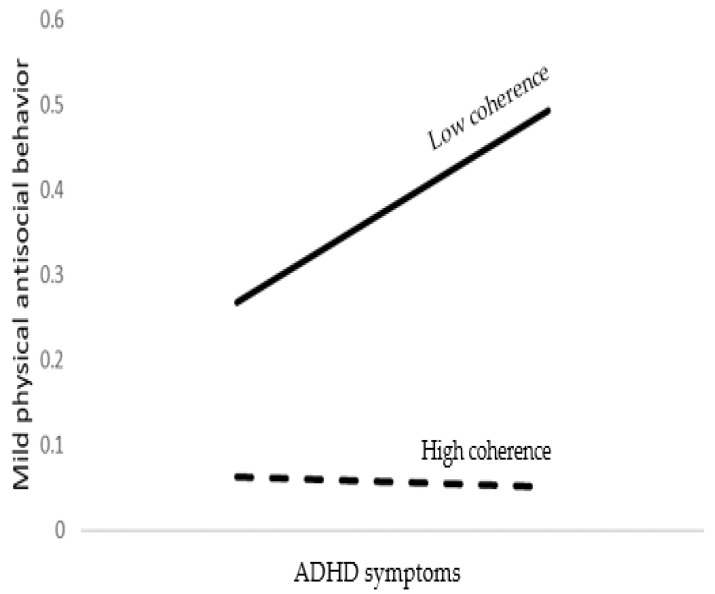
The association between ADHD symptoms and mild physical antisocial behavior as moderated by coherence.

**Figure 3 brainsci-12-01336-f003:**
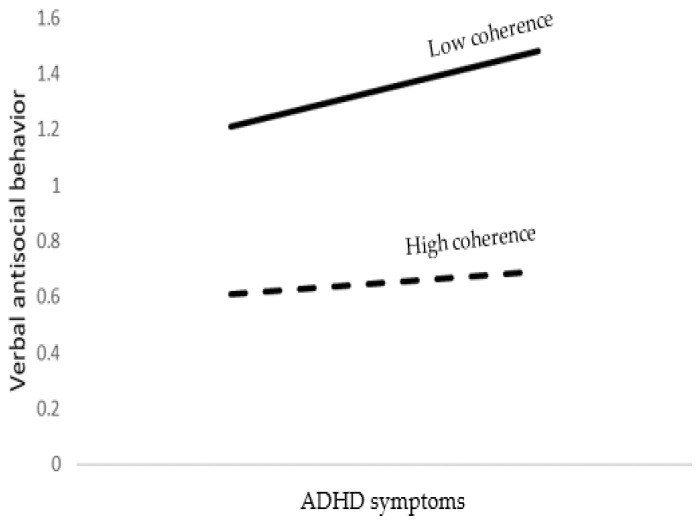
The association between ADHD symptoms and verbal antisocial behavior as moderated by coherence.

**Figure 4 brainsci-12-01336-f004:**
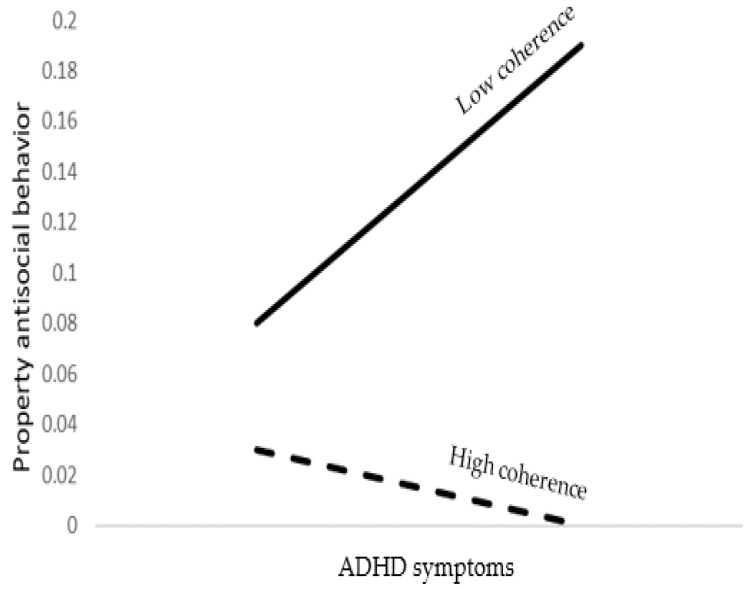
The association between ADHD symptoms and property crimes as moderated by coherence.

**Figure 5 brainsci-12-01336-f005:**
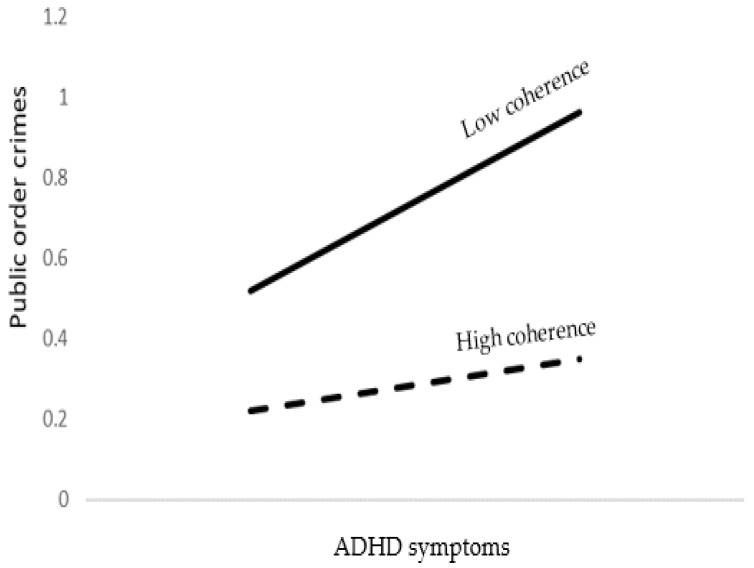
The association between ADHD symptoms and public order crimes as moderated by coherence.

**Figure 6 brainsci-12-01336-f006:**
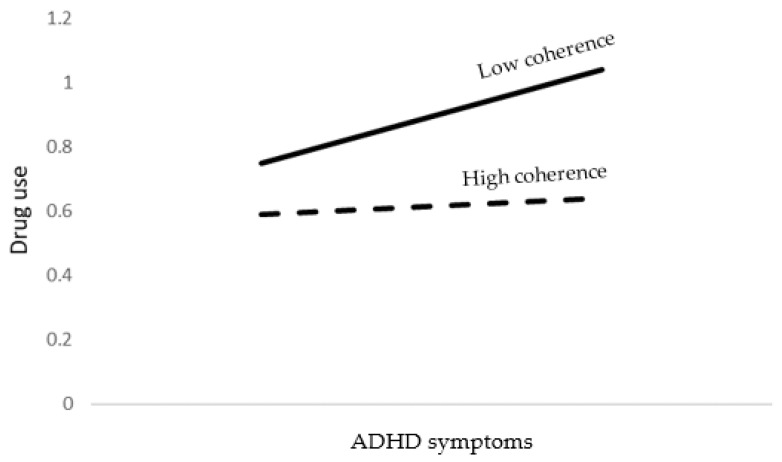
The association between ADHD symptoms and drug use as moderated by coherence.

**Table 1 brainsci-12-01336-t001:** Descriptive statistics of the demographic characteristics, ADHD, and antisocial behavior types.

	N	Median	Mean	SD	Minimum	Maximum
Religiosity	3165	3.00	2.76	1.44	0.00	4.00
Emotional symptoms	2804	1.40	1.56	0.48	1.00	3.00
ADHD symptoms	3179	1.28	1.37	0.72	0.00	4.00
Coherence	2963	4.92	4.78	1.16	1.00	7.00
Severe physical antisocial behavior	3079	0.00	0.11	0.35	0.00	4.00
Mild physical antisocial behavior	3079	0.00	0.24	0.52	0.00	4.00
Verbal antisocial behavior	3079	0.75	1.01	0.93	0.00	4.00
Property crimes	3079	0.00	0.09	0.33	0.00	4.00
Public order crimes	3070	0.33	0.51	0.76	0.00	4.00
Drug use	2834	0.40	0.77	1.13	0.00	7.00

**Table 2 brainsci-12-01336-t002:** Nonparametric correlations between ADHD, antisocial behavior types, and coherence.

	ADHD	Severe Physical Antisocial Behavior	Mild Physical Antisocial Behavior	Verbal Antisocial Behavior	Property Crimes	Public Order Crimes	Drug Use	Coherence
ADHD symptoms	1.000							
Severe physical antisocial behavior	0.175 **	1.000						
Mild physical antisocial behavior	0.207 **	0.542 **	1.000					
Verbal antisocial behavior	0.234 **	0.403 **	0.520 **	1.000				
Property crimes	0.176 **	0.459 **	0.418 **	0.365 **	1.000			
Public order crimes	0.246 **	0.335 **	0.382 **	0.464 **	0.346 **	1.000		
Drug use	0.117 **	0.130 **	0.143 **	0.203 **	0.186 **	0.254 **	1.000	
Coherence	−0.424 **	−0.195 **	−0.283 **	−0.363 **	−0.259 **	−0.288 **	−0.086 **	1.000

**. correlation is significant at the 0.01 level (2-tailed).

**Table 3 brainsci-12-01336-t003:** ADHD as a predictor of severe physical antisocial behavior moderated by coherence.

	Beta	B	S.E.	t	P	95%CI	
						Lower	Upper
ADHD symptoms	0.127	0.062	0.009	6.79	<0.001	0.041	0.083
Coherence	−0.190	−0.057	0.007	−8.51	<0.001	−0.077	−0.043
ADHD symptoms × Coherence	−0.172	−0.061	0.006	−10.18	<0.001	−0.089	−0.034
Age	−0.159	−0.143	0.015	−9.48	<0.001	−0.191	−0.102
Gender	−0.094	−0.066	0.012	−5.58	<0.001	−0.089	−0.039
Religiosity	−0.006	−0.002	0.004	−0.37	0.711	−0.010	0.007
Emotional symptoms	−0.105	−0.076	0.016	−4.87	<0.001	−0.122	−0.041
	Conditional effects of ADHD symptoms						
Low coherence	0.299	0.133	0.011	12.76	<0.001	0.113	0.154
High coherence	−0.045	−0.009	0.012	−0.68	0.491	−0.033	0.016

**Table 4 brainsci-12-01336-t004:** ADHD symptoms as a predictor of mild physical antisocial behavior moderated by coherence.

	Beta	B	S.E.	t	P	95%CI	
						Lower	Upper
ADHD symptoms	0.102	0.074	0.013	5.57	<0.001	0.049	0.103
Coherence	−0.311	−0.138	0.010	−14.28	<0.001	−0.171	−0.117
ADHD symptoms × Coherence	−0.135	−0.070	0.009	−8.17	<0.001	−0.098	−0.044
Age	−0.186	−0.247	0.022	−11.34	<0.001	−0.311	−0.183
Gender	−0.099	−0.103	0.017	−6.00	<0.001	−0.139	−0.073
Religiosity	0.056	0.020	0.006	3.35	<0.001	0.011	0.038
Emotional symptoms	−0.109	−0.116	0.022	−5.19	<0.001	−0.171	−0.069
	Conditional effects of ADHD symptoms						
Low coherence	0.237	0.156	0.015	10.35	<0.001	0.126	0.185
High coherence	−0.033	−0.008	0.018	−0.46	0.643	−0.044	0.027

**Table 5 brainsci-12-01336-t005:** ADHD symptoms as a predictor of verbal antisocial behavior moderated by coherence.

	Beta	B	S.E.	t	P	95%CI	
						Lower	Upper
ADHD symptoms	0.098	0.128	0.023	5.46	<0.001	0.088	0.168
Coherence	−0.371	−0.297	0.017	−17.31	<0.001	−0.334	−0.264
ADHD symptoms × Coherence	−0.059	−0.055	0.015	−3.65	<0.001	−0.088	−0.029
Age	−0.159	−0.380	0.038	−9.89	<0.001	−0.468	−0.278
Gender	−0.113	−0.211	0.030	−6.97	<0.001	−0.285	−0.156
Religiosity	−0.051	−0.033	0.011	−3.11	0.002	−0.053	−0.014
Emotional symptoms	−0.108	−0.207	0.040	−5.23	<0.001	−0.296	−0.127
	Conditional effects of ADHD symptoms						
Low coherence	0.157	0.19	0.027	7.23	<0.001	0.140	0.245
High coherence	0.039	0.63	0.032	1.97	0.048	0.001	0.126

**Table 6 brainsci-12-01336-t006:** ADHD as a predictor of property crimes moderated by coherence.

	Beta	B	S.E.	t	P	95%CI	
						Lower	Upper
ADHD symptoms	0.063	0.029	0.009	3.29	0.001	0.001	0.046
Coherence	−0.190	−0.053	0.006	−8.34	<0.001	−0.069	−0.040
ADHD symptoms × Coherence	−0.135	−0.044	0.006	−7.78	<0.001	−0.079	−0.022
Age	−0.109	−0.091	0.014	−6.34	<0.001	−0.134	−0.049
Gender	−0.112	−0.073	0.011	−6.50	<0.001	−0.104	−0.049
Religiosity	−0.081	−0.018	0.004	−4.62	<0.001	−0.027	−0.009
Emotional symptoms	−0.065	−0.043	0.015	−2.94	0.003	−0.073	−0.016
	Conditional effects of ADHD symptoms						
Low coherence	0.198	0.080	0.010	8.08	<0.001	0.061	0.099
High coherence	−0.072	−0.023	0.012	−1.91	0.055	−0.046	0.001

**Table 7 brainsci-12-01336-t007:** ADHD symptoms as a predictor of public order antisocial behavior moderated by coherence.

	Beta	B	S.E.	t	P	95%CI	
						Lower	Upper
ADHD symptoms	0.166	0.176	0.019	9.09	<0.001	0.135	0.212
Coherence	−0.279	−0.182	0.014	−12.84	<0.001	−0.213	−0.145
ADHD symptoms × Coherence	−0.099	−0.075	0.013	−5.99	<0.001	−0.104	−0.046
Age	−0.164	−0.319	0.032	−10.05	<0.001	−0.404	−0.251
Gender	−0.072	−0.110	0.025	−4.41	<0.001	−0.159	−0.064
Religiosity	−0.096	−0.051	0.009	−5.75	<0.001	−0.070	−0.033
Emotional symptoms	−0.154	−0.241	0.033	−7.36	<0.001	−0.304	−0.174
	Conditional effects of ADHD symptoms						
Low coherence	0.265	0.264	0.022	11.99	<0.001	0.221	0.307
High coherence	0.067	0.088	0.026	3.33	<0.001	0.036	0.139

**Table 8 brainsci-12-01336-t008:** ADHD symptoms as a predictor of drug use moderated by coherence.

	Beta	B	S.E.	t	P	95%CI	
						Lower	Upper
ADHD symptoms	0.075	0.119	0.029	4.09	<0.001	0.070	0.188
Coherence	−0.122	−0.118	0.021	−5.55	<0.001	−0.165	−0.074
ADHD symptoms × Coherence	−0.062	−0.071	0.019	−3.73	<0.001	−0.118	−0.023
Age	0.092	0.267	0.048	5.59	<0.001	0.166	0.386
Gender	−0.323	−0.732	0.038	−19.43	<0.001	−0.814	−0.656
Religiosity	−0.235	−0.185	0.013	−13.96	<0.001	−0.216	−0.156
Emotional symptoms	−0.067	−0.156	0.049	−3.17	0.002	−0.261	−0.060
	Conditional effects of ADHD symptoms						
Low coherence	0.137	0.201	0.033	6.08	<0.001	0.137	0.266
High coherence	0.013	0.036	0.040	0.92	0.355	−0.041	0.115

## Data Availability

The data presented in this study are available on request from the corresponding author. The data are not publicly available because they contain self-reported criminal behavior.

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
