# Peer review of "The Link between ADHD Symptoms and Antisocial Behavior: The Moderating Role of the Protective Factor Sense of Coherence"

_brainsci, 2022, doi:10.3390/brainsci12101336_

Round 1

Reviewer 1 Report

Thank you for providing me with the opportunity to review the article on how sense of coherence moderates the link between ADHD and antisocial behavior.

The sample size is huge which is a particular strength for the current study.

The authors should provide further details on how the ‘convenient sampling’ was conducted. Where did the link for the questionnaire was posted exactly? How recruitment was achieved. It is indeed a massive limitation that 455 questionnaires were excluded. That opens the door widely to selection bias, people with ADHD are more likely not to finish the questionnaire. There are many ways to deal with missing data such as MICE rather than exclusion. Please reanalyze the data after imputation of missing data.

Adult ADHD Self-Report Scale is notorious for poor specificity. The authors are encouraged to mention that in the limitation section of the paper.

The authors stated that ‘antisocial behavior [was regarded] as the predicted variable’ however it is unclear what was its distribution. Did they conduct a test of normality? Was the regression modelling a Poisson one or a negative binomial? Please provided further details on the type of modelling used.

A far better approach than ‘further exploratory analysis of the ADHD-Antisocial behavior interaction’ is the so-called ‘moderation analysis’ using the concept of Structured Equation Modelling approach.

Table 2 shows results of ‘Nonparametric Correlations between ADHD, antisocial behavior types and coherence’. However, I note that the authors repeatedly tested for p values for Spearman’s correlation 49 times without allowing for correctional methods that deflate the chance of type 1 error (such as Bonferroni method for instance). Please make necessary adjustments and re-show the results.

What is shown in table 3 is NOT ‘moderation analysis’. It is actually an addition of an interaction term between ADHD and SOC into the overall ‘presumably’ linear regression model that predicts antisocial behavior. Please conduct proper moderation analysis using the concept of Structured Equation Modelling approach and display its results.

I did not like the idea of breaking down antisocial behavior into multiple sub-behaviors and running multiple models (as they show in tables 3-6) without statistical correction. The authors have should either run a single regression model for the overall antisocial behavior or include all sub-behaviors in a SEM with multiple predicted variables.

Author Response

Thank you for providing me with the opportunity to review the article on how sense of coherence moderates the link between ADHD and antisocial behavior.

The sample size is huge which is a particular strength for the current study.

Response: We are thankful for the thorough review. Your comments helped us crystalize essential issues and deepen and establish our research. 

The authors should provide further details on how the ‘convenient sampling’ was conducted. Where did the link for the questionnaire was posted exactly? How recruitment was achieved.

Response: Following this comment, we inserted the next explanation in the methods section (p. 11, line 127):

The participants were recruited by convenience sampling, using an online questionnaire distributed on social networks (Whatsapp, Instagram, Facebook, and emails) in Israel in October-November 2020. We explained that our research goal was to understand individuals’ thoughts, feelings, behaviors, and coping strategies in various life situations. For those interested in participating, we offered a lottery of purchase tickets in a separate link (the winning ratio was 1:50, and the ticket value was about 12$). We also asked the participants to answer the questionnaires and further distribute it Three thousand four hundred fifteen participants between the ages of 15 and 50 responded to the questionnaire. Two hundred thirty-five participants were excluded since they did not answer any questionnaires or answered the questionnaires randomly., resulting in Three thousand one hundred eighty that were included in the statistical analyses.

 It is indeed a massive limitation that 455 questionnaires were excluded. That opens the door widely to selection bias, people with ADHD are more likely not to finish the questionnaire. There are many ways to deal with missing data such as MICE rather than exclusion. Please reanalyze the data after imputation of missing data.

Response: Following this important comment, we returned to the sampling 220 participants who had partial answers (see p. 3, line 127). The rest of the participants either did not reply to even one of the questionnaires, or replied randomly. In addition, as mentioned below, we reanalyzed the data using a SEM model, which included missing values analysis.

Adult ADHD Self-Report Scale is notorious for poor specificity. The authors are encouraged to mention that in the limitation section of the paper.

Response: Regarding this comment, we addressed this issue in the limitation section. (p. 19, lines 526):

Furthermore, in some studies, Adult ADHD Self-Report Scale had low specificity [58, 59]. It is important to note that the research focuses on ADHD symptoms in the general population and not on diagnosed ADHD. Further research is needed to determine whether these correlations exist also in diagnosed population.

The authors stated that ‘antisocial behavior [was regarded] as the predicted variable’ however it is unclear what was its distribution. Did they conduct a test of normality? Was the regression modelling a Poisson one or a negative binomial? Please provided further details on the type of modelling used.

A far better approach than ‘further exploratory analysis of the ADHD-Antisocial behavior interaction’ is the so-called ‘moderation analysis’ using the concept of Structured Equation Modelling approach.

Response: As common for antisocial behavior scales, the distributions did depart from normality, though the residuals were not seriously skewed. To account for potential bias due to non-normality, we estimated parameters’ 95% confidence intervals using bootstrapping methods. The bootstrapping technique is more robust to non-normality than the standard p-value. It is important to note that both the p-value approach and bootstrapping yielded similar results. Moreover, the large sample size in the current investigation also minimizes the potential bias of non-normality (e.g., Bishara, A. J., & Hittner, J. B., 2012). Further, following the reviewer’s suggestion, we applied a Structured Equation Modelling (SEM) approach to examine the moderation hypothesis.

Bishara, A. J., & Hittner, J. B. (2012). Testing the significance of a correlation with nonnormal data: comparison of Pearson, Spearman, transformation, and resampling approaches. Psychological methods, 17(3), 399.

Table 2 shows results of ‘Nonparametric Correlations between ADHD, antisocial behavior types and coherence’. However, I note that the authors repeatedly tested for p values for Spearman’s correlation 49 times without allowing for correctional methods that deflate the chance of type 1 error (such as Bonferroni method for instance). Please make necessary adjustments and re-show the results.

Response: In response to this important comment, we allowed for Bonferroni corrections (p = 0.05/49 = 0.00102). All the bivariate correlations in Table 2 met this criterion. This change is now mentioned in the manusciprt (p. 6, line 250).

What is shown in table 3 is NOT ‘moderation analysis’. It is actually an addition of an interaction term between ADHD and SOC into the overall ‘presumably’ linear regression model that predicts antisocial behavior. Please conduct proper moderation analysis using the concept of Structured Equation Modelling approach and display its results.

I did not like the idea of breaking down antisocial behavior into multiple sub-behaviors and running multiple models (as they show in tables 3-6) without statistical correction. The authors have should either run a single regression model for the overall antisocial behavior or include all sub-behaviors in a SEM with multiple predicted variables.

Response: Following your important comments, we conducted a moderation analysis in SEM using AMOS 26. We included all 6 facets of antisocial behavior as dependent variables in the tested model. A detailed description of the SEM model appears in the analytical strategy section (p. 5, line 200).

Reviewer 2 Report

Dear Authors; I fond this study on the moderation effect of SOC on the relationship between ADHD and ANTSB very interesting in terms of topic and statistical analysis. Given its huge volume of done work, it was natural to have some minor issues needed attention before publication. Regards. P.S.

[1] Writing:

1-1 Make sure references are in MDPI format. For example years for articles are in bold font. etc.

1-2 Add list of Abbreviation used in the work right before references section for readers referral.

1-3 In lines 17-21 in abstract, add effect estimates and p-values.

1-4 In line 198: SPSS27.0 needs its own citation:

*IBM Corp. Released 2020. IBM SPSS Statistics for Windows, Version 27.0. Armonk, NY: IBM Corp

[2] Statistical:

2-1 Section "3.3 Moderation Analysis": Write down your statistical model formulae in the text .

2-2 Correlations: In section 3.2 you report on Spearman correlations but no Pearson correlations. Did you try Pearson correlations ?  Explain ? I would like to see a Table reporting them in the  supplementary materials and have one paragraph in the text comparing them with those of Spearman ones. 

2-3 Figures 1,2,3,4,5,6: Add p-values in the caption. the readers need to see the size of significancy !

Author Response

Dear Authors; I found this study on the moderation effect of SOC on the relationship between ADHD and ANTSB very interesting in terms of topic and statistical analysis. Given its huge volume of done work, it was natural to have some minor issues needed attention before publication. Regards. P.S.

Response: We thank the reviewer for the positive impression.

[1] Writing:  

1-1 Make sure references are in MDPI format. For example years for articles are in bold font. etc.

Response: We adjusted the style to the MDPI style.

1-2 Add list of Abbreviation used in the work right before references section for readers referral.

Response: We added a list of Abbreviations used in the work before the references section.

1-3 In lines 17-21 in abstract, add effect estimates and p-values.

Response: Effecrt estimates and p value were added to the abstract.

1-4 In line 198: SPSS27.0 needs its own citation:

Response: In line with reviewer #1 suggestion that demanded changing the statistical analyses, we used AMOS instead of SPSS and added a reference accordingly.

[2] Statistical:  

2-1 Section “3.3 Moderation Analysis”: Write down your statistical model formulae in the text .

Response: In line with reviewer #1 suggestion, we reanalyzed the data using a SEM approach (see the analytical strategy for a detailed description). Unlike regression analysis, which runs a single equation, SEM runs multiple equations simultaneously with a maximum-likelihood estimation. Describing the mathematical equations may seem overwhelming, at least in our view, to the common reader. However, if the Editor and reviewer believe that this description is necessary for a  comprehensive understanding of the statistical analyses, we are willing to add it to the manuscript.

2-2 Correlations: In section 3.2 you report on Spearman correlations but no Pearson correlations. Did you try Pearson correlations ? Explain ? I would like to see a Table reporting them in the supplementary materials and have one paragraph in the text comparing them with those of Spearman ones.

Response: We calculated Pearson correlations between the research variables. The correlation matrix appears in the supplementary materials. All the Pearson correlations were statistically significant and in the same direction as were the Spearman correlations. Additionally, the Pearson and Spearman correlations were very similar. In fact, for 20 out of the 28 correlations, the difference in magnitude between Pearson and Spearman ranged between 0 to 0.1, for the rest the differences ranged between 0.11 to 0.19. Thus, overall, the correlation matrices were fairly similar.

2-3 Figures 1,2,3,4,5,6: Add p-values in the caption. the readers need to see the size of significant!

Response: Thank you for this suggestion. We realize that we did not explicitly report the conditional effect in the previous version of the manuscript. In the revised manuscript, we describe in detail the significance of the conditional effects along with its standardized estimation and 95%CI in the Table that accompanied the Figure. To minimize repetition, we did not include the same data in the Figure.